# Incorporation of multiple supramolecular binding sites into a robust MOF for benchmark one-step ethylene purification

Enyu Wu [1,4], Xiao-Wen Gu[1,4], Di Liu[1], Xu Zhang [2], Hui Wu [3], Wei Zhou [3], Guodong Qian[1] & Bin Li [1] ✉

One-step adsorption separation of $C_2H_4$ from ternary $C_2$ hydrocarbon mixtures remains an important and challenging goal for petrochemical industry. Current physisorbents either suffer from unsatisfied separation performance, poor stability, or are difficult to scale up. Herein, we report a strategy of constructing multiple supramolecular binding sites in a robust and scalable MOF (Al-PyDC) for highly efficient one-step $C_2H_4$ purification from ternary mixtures. Owing to suitable pore confinement with multiple supramolecular binding sites, Al-PyDC exhibits one of the highest $C_2H_2$ and $C_2H_6$ uptakes and selectivities over $C_2H_4$ at ambient conditions. The gas binding sites have been visualized by single-crystal X-ray diffraction studies, unveiling that the low-polarity pore surfaces with abundant electronegative N/O sites provide stronger multiple supramolecular interactions with $C_2H_2$ and $C_2H_6$ over $C_2H_4$. Breakthrough experiments showed that polymer-grade $C_2H_4$ can be separated from ternary mixtures with a maximum productivity of 1.61 mmol g$^{-1}$. This material can be prepared from two simple reagents using a green synthesis method with water as the sole solvent, and its synthesis can be easily scaled to multikilogram batches. Al-PyDC achieves an effective combination of benchmark separation performance, high stability/recyclability, green synthesis and easy scalability to address major challenges for industrial one-step $C_2H_4$ purification.

Purification of olefins such as ethylene ($C_2H_4$) accounts for 0.3% of global energy consumption, highlighting as one of seven most important chemical separations[1]. As the largest-volume product of the chemical industry, the worldwide production of $C_2H_4$ has exceeded 200 million tons per year in 2020 and will continuously increase in the future. At present, polymer-grade $C_2H_4$ is mainly produced by energy-intensive separation of downstream $C_2$ hydrocarbon gas mixtures during the steam cracking process[2]. Acetylene ($C_2H_2$) is first removed through catalytic hydrogenation using noble-metal catalysts under high temperatures and pressures, and then ethane ($C_2H_6$) is separated by energy-intensive cryogenic distillation[3]. These high energy footprints associated with $C_2H_4$ purification have pushed the development of cost- and energy-efficient separation technologies to a level of utmost importance.

Adsorption separation technology based on porous materials has been demonstrated to be a promising technology to replace traditional cryogenic distillation and thus to fulfill the energy-efficient separation economy[4–9]. Highly efficient separation of $C_2H_4$ from binary

[1]State Key Laboratory of Silicon and Advanced Semiconductor Materials, School of Materials Science and Engineering, Zhejiang University, Hangzhou 310027, China. [2]School of Chemistry and Chemical Engineering, Huaiyin Normal University, Huaian 223300, China. [3]NIST Center for Neutron Research, National Institute of Standards and Technology, Gaithersburg, MD 20899-6102, USA. [4]These authors contributed equally: Enyu Wu, Xiao-Wen Gu. ✉e-mail: bin.li@zju.edu.cn

$C_2H_2/C_2H_4$ and $C_2H_6/C_2H_4$ mixtures has been well addressed by various materials including metal–organic frameworks (MOFs)[10–19], covalent organic frameworks (COFs)[20], zeolites and so on[21–23]. Amongst them, microporous MOFs have shown particular promise for gas separations because of the powerful tunability on pore size and functionality[24–30]. Compared with separation of binary mixtures, simultaneous removal of $C_2H_2$ and $C_2H_6$ from ternary $C_2$ mixtures would be more desirable to directly obtain polymer-grade $C_2H_4$, which can simplify the separation process to result in large energy saving. However, the simultaneous separation of $C_2H_2$ and $C_2H_6$ impurities from $C_2H_4$ remains a daunting challenge for classical physisorbents, since all the physical properties of $C_2H_4$ molecule lie between $C_2H_2$ and $C_2H_6$ (Supplementary Table 1)[31, 32]. Owing to the higher quadrupole moment and acidity of $C_2H_2$ over $C_2H_4$, the preferential adsorption of $C_2H_2$ over $C_2H_4$ with high selectivities can be generally achieved by MOFs with highly polar groups (e.g., open metal sites and fluoridated anion pillars)[33–41]; however, these kinds of materials bind more strongly with $C_2H_4$ over $C_2H_6$ to result in the selective adsorption of $C_2H_4$ over $C_2H_6$ (Fig. 1). Given that $C_2H_6$ has a higher polarizability than $C_2H_4$ ($44.7 \times 10^{-25}$ vs $42.52 \times 10^{-25}$ cm$^3$), the selective adsorption of $C_2H_6$ over $C_2H_4$ is commonly favored by MOFs with nonpolar/inert pore surfaces (e.g., aromatic or aliphatic moieties), wherein soft supramolecular interactions (e.g., C−H···π or hydrogen bonding) make major contributions[42–48]. A handful of $C_2H_6$-selective MOFs have been recently discovered to show the preferential adsorption of both $C_2H_2$ and $C_2H_6$ over $C_2H_4$ via various supramolecular interactions[49–59]. However, the weak nature of supramolecular interactions commonly makes the $C_2H_2$ or $C_2H_6$ binding affinity not so sufficient, resulting in poor gas uptake or insufficient selectivity to preclude most of them from being highly selective (Fig. 1). For example, Azole-Th-1 and Zn(ad)(int) exhibit relatively high $C_2H_6/C_2H_4$ selectivity, whereas their $C_2H_2$ uptake and selectivity are quite low because of their insufficient $C_2H_2$ binding affinity[49,54]. UiO-67-(NH$_2$)$_2$ and CuTiF$_6$-TPPY holds the benchmark $C_2H_2$ and $C_2H_6$ uptakes or selectivities, but limited by inadequate gas selectivities or uptakes[53,59], respectively. Ideal adsorbents for one-step $C_2$ separation should possess both high $C_2H_2$ and $C_2H_6$ adsorption capacities and selectivities over $C_2H_4$, which can maximize the productivity and purity of $C_2H_4$ purification. Besides separation performance, some other factors such as stability, economy feasibility, and synthesis scalability, also need to be considered for large-scale industrial applications. However, most of the reported $C_2H_2/C_2H_6$-selective

MOFs suffer from the drawbacks of poor water stability, high cost, or low scalability, largely impeding their applications in industrial scenarios. Currently, there are no reports existing on kilogram-scale synthesis of MOFs relevant for one-step $C_2H_2/C_2H_6/C_2H_4$ separation. Developing ideal MOF adsorbents that fully merge high separation performance with high stability, economy feasibility and easy scalability of synthesis has never been achieved yet for this important one-step $C_2H_4$ purification.

To overcome the weakness nature of supramolecular interactions, an effective design strategy is to incorporate multiple supramolecular binding sites into MOFs with suitable pore sizes to enforce $C_2H_2$ and $C_2H_6$ binding affinity (Fig. 1), thus concurrently improving their uptake and selectivity over $C_2H_4$. Recent studies have shown that the incorporation of electronegative oxygen or nitrogen sites into MOFs can provide hydrogen-bonding interactions with acidic $C_2H_2$, and thus enhance $C_2H_2$ adsorption and separation capacities[59–63]. Further, such O/N binding sites may also enable the formation of more numbers of supramolecular interactions with $C_2H_6$ molecule than with $C_2H_4$, given that $C_2H_6$ has larger molecule size and more H atoms[16, 64]. With the above considerations in mind, we herein report a strategy of designing multiple supramolecular binding sites in microporous MOFs for highly efficient one-step separation of $C_2H_4$ from ternary $C_2$ mixtures. We target this matter in a known and robust Al-based MOF, [Al(OH)PyDC]$_n$ (Al-PyDC, also called KMF-1 or MOF-313, H$_2$PyDC = 2,5-pyrroledicarboxylate)[65,66]. This material consists of cis-corner-sharing octahedra AlO$_6$ chains linked by N-containing aromatic ligands to form one-dimensional pore channels with a suitable size of 5.8 Å, wherein a large number of polar O sites and N-heterocyclic rings are densely arranged along pore channels to provide abundant supramolecular binding sites (Fig. 2). Al-PyDC thus exhibits one of the highest $C_2H_2$ and $C_2H_6$ uptakes (8.24 and 4.20 mmol g$^{-1}$) and selectivities (4.3 and 1.9) over $C_2H_4$ at ambient conditions, outperforming most benchmark materials reported so far. Single-crystal X-ray diffraction (SCXRD) studies on gas-loaded Al-PyDC for all the $C_2$ molecules visually unveiled that multiple O/N sites on channel-pore surfaces provide stronger multiple supramolecular interactions with both $C_2H_2$ and $C_2H_6$ over $C_2H_4$, accounting for both very high $C_2H_2$ and $C_2H_6$ uptakes and selectivities. Dynamic breakthrough experiments confirm its exceptional one-step $C_2H_4$ purification from 1/49.5/49.5 or 1/9/90 $C_2H_2/C_2H_6/C_2H_4$ mixtures at ambient conditions, affording the maximal polymer-grade $C_2H_4$ productivity of 0.66 or 1.61 mmol g$^{-1}$. Most importantly, Al-PyDC was synthesized from two simple and commercially available reagents H$_2$PyDC and AlCl$_3$·6H$_2$O using water as the sole solvent, and we showed that its synthesis can be easily scaled-up to multikilogram batches in a high yield of 92% at mild and water-based conditions. The production of Al-PyDC can be considered as a scalable green synthesis. The combined advantages of benchmark separation performance, high stability/recyclability, green synthesis and easy scalability make it as a benchmark material for industrial one-step $C_2H_2/C_2H_6/C_2H_4$ separation.

## Results

### Synthesis and structural characterization
The Al-PyDC samples were readily synthesized as microcrystalline powders by reaction of the simple dicarboxylate ligand (H$_2$PyDC) and AlCl$_3$·6H$_2$O according to the previous literature with procedural optimization (see the Method for details)[65]. The crystal structures of the as-synthesized and activated Al-PyDC have been determined by Chang et al. based on the synchrotron powder diffraction data[65]. In spite of this, it is still highly important to obtain large enough single crystals of Al-PyDC for SCXRD analysis, which facilitates to not only check the accurate structure information but also determine gas-loaded crystal structures to visually identify the binding sites of $C_2$ molecules. While it is very challenging to get large single crystals for Al-MOFs, we here successfully prepared large colorless square-block-shaped crystals for

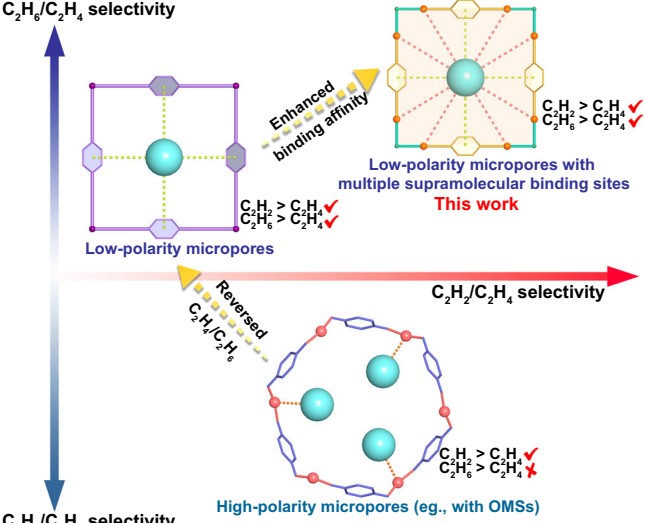

**Fig. 1 | Illustration of strategy.** Schematic illustration of the proposed multiple supramolecular binding sites for boosting the selective binding of $C_2H_2$ and $C_2H_6$ over $C_2H_4$ molecule.

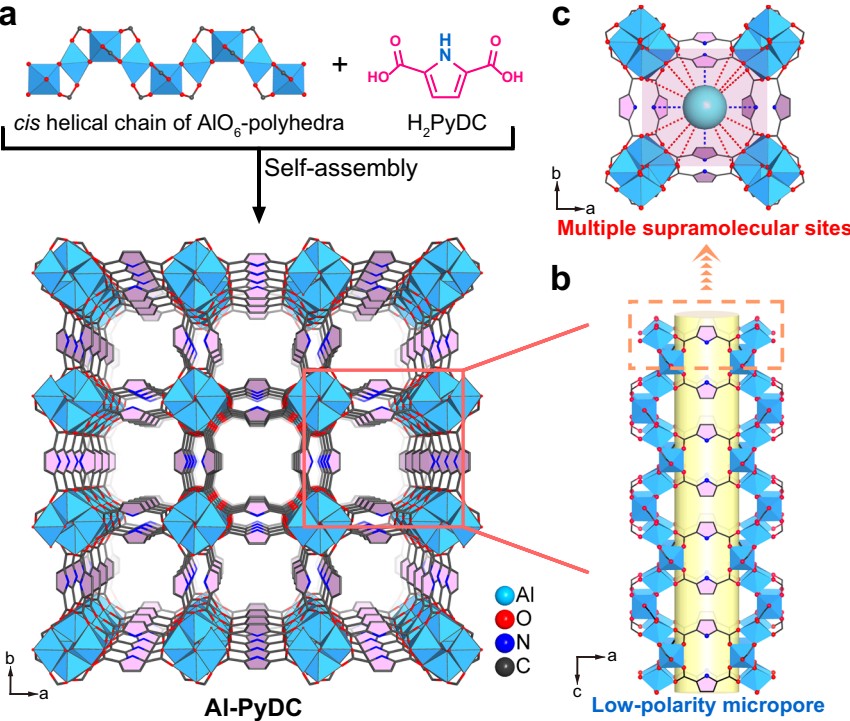

**Fig. 2 | Crystal structure description of the activated Al-PyDC. a** The framework of Al-PyDC formed by AlO$_6$-polyhedra chains and H$_2$PyDC linkers. **b** The 1D square channel with low-polarity micropore surfaces, decorated by abundant O/N sites and aromatic rings. **c** Multiple supramolecular binding sites in the pore of Al-PyDC.

Al-PyDC through great endeavors to optimize the growth method (Supplementary Fig. 1). The SCXRD analysis revealed that hydrated Al-PyDC crystallizes in a non-centrosymmetric tetragonal space group (*I*4$_1$md) with unit cell parameters of *a* = *b* = 21.1895(3) Å and *c* = 10.6283(3) Å, which are consistent with the results obtained by synchrotron powder diffraction data. After removing all the guest molecules, the activated structure of Al-PyDC can transfer to a new tetragonal symmetry of *I*4$_1$/amd, as evidenced by the literature and our following gas-loaded crystal structures (Supplementary Table 4).

The crystal structure of the activated Al-PyDC is depicted in Fig. 2. Each octahedrally coordinated Al-centre is linked to four oxygen atoms from four PyDC linkers and two *cis* bridging OH$^-$ groups. The octahedral AlO$_6$-polyhedras form one-dimensional (1D) four-fold helical chains via corner sharing, which are further interconnected with each other by the PyDC linkers to result in a 3D framework (Fig. 2a). As shown in Fig. 2b, Al-PyDC exhibits a 1D square-shaped pore channel with a pore window size of 5.8 Å in diameter along the *c* axis. This aperture size is larger than the kinetic diameter of all C$_2$ molecules so as to favor the rapid diffusion of these gas molecules. Due to the full coordination of the Al-centre, there are no polar open metal sites (OMSs) existed in the framework of Al-PyDC. Most importantly, the surface of pore channels is surrounded by abundant naked oxygen atoms and N-containing aromatic rings originated from organic linkers and bridging OH$^-$ groups. Such nonpolar pore surface decorated by abundant O/N sites and aromatic rings may provide multiple supramolecular binding sites to preferentially interact with C$_2$H$_6$ than with C$_2$H$_4$ molecule (Fig. 2c). Further, recent studies on Al-MOFs have shown that the densely distributed oxygen atoms around the channels can provide high-density H-bonding interactions with C$_2$H$_2$ molecule, leading to high C$_2$H$_2$ uptakes and selectivities[61,62]. Therefore, we reasoned that such multiple supramolecular binding sites within OMS-free Al-PyDC may show the great potential to not only concurrently reinforce C$_2$H$_2$ and C$_2$H$_6$ binding affinity for high gas uptakes, but also provide more preferential adsorption of C$_2$H$_2$ and C$_2$H$_6$ over C$_2$H$_4$ for high selectivities.

## Gas adsorption measurements

The solvent-exchanged Al-PyDC sample was evacuated at room temperature for 12 h and then 393 K for 12 h until the outgas rate was 5 μmHg min$^{-1}$, affording the fully activated material (Supplementary Fig. 2). The permanent porosity of the activated Al-PyDC was determined by nitrogen (N$_2$) adsorption isotherms at 77 K. As shown in Fig. 3a, Al-PyDC takes up a 305 cm$^3$ g$^{-1}$ amount of N$_2$ at 77 K and 1 bar, with a significant type I sorption behavior. The Brunauer−Emmett−Teller surface area and pore volume were calculated to be 1134 m$^2$ g$^{-1}$ and 0.472 cm$^3$ g$^{-1}$ (Supplementary Fig. 3), in good agreement with the values (1130 m$^2$ g$^{-1}$ and 0.473 cm$^3$ g$^{-1}$) reported in the literature[65]. The pore size distribution, determined by Non-Local Density Functional Theory (NLDFT) method based on 77 K N$_2$ isotherms, shows a moderate pore size of 5.8 Å (Fig. 3a), which is consistent well with the value obtained from the crystal structure.

Pure-component adsorption isotherms of C$_2$ hydrocarbons for Al-PyDC were measured at 273, 296, and 313 K up to 1 bar (Fig. 3b, Supplementary Figs. 4 and 5), respectively. Owing to the OMS-free and nonpolar pore surfaces, Al-PyDC exhibits an obviously preferential adsorption of C$_2$H$_2$ and C$_2$H$_6$ over C$_2$H$_4$ at 296 K (Fig. 3b). The C$_2$ gas uptakes at 296 K and 1 bar follow the expected sequence of C$_2$H$_2$ (8.24 mmol g$^{-1}$) > C$_2$H$_6$ (4.20 mmol g$^{-1}$) > C$_2$H$_4$ (3.44 mmol g$^{-1}$). This endows Al-PyDC with a promising capacity for efficient one-step separation of C$_2$H$_4$ from ternary mixtures. It is worthy to note that the C$_2$H$_2$ uptake (8.24 mmol g$^{-1}$) at 1 bar and 296 K is the highest among all the C$_2$H$_2$/C$_2$H$_6$-selective MOFs reported so far (Fig. 3d and Supplementary Fig. 6), notably larger than that of most benchmark materials such as CuTiF$_6$-TPPY (3.62 mmol g$^{-1}$)[53], TJT-100 (4.46 mmol g$^{-1}$)[55], NPU-1 (5.09 mmol g$^{-1}$)[56] and UiO-67-NH$_2$ (5.90 mmol g$^{-1}$)[59]. Further, the C$_2$H$_6$ uptake of Al-PyDC is also among the top-tier values for the relevant MOFs (Fig. 3e), surpassing that of Ag-PCM-102 (3.70 mmol g$^{-1}$)[52], CuTiF$_6$-TPPY (2.82 mmol g$^{-1}$)[53], TJT-100 (3.75 mmol g$^{-1}$)[55] and comparable to Azole-Th-1 (4.47 mmol g$^{-1}$)[49] and NPU-1 (4.50 mmol g$^{-1}$)[56]. Evidently, both the top-tier C$_2$H$_2$ and C$_2$H$_6$ uptakes make Al-PyDC as one of the best materials for ternary C$_2$ separation. Further, we also

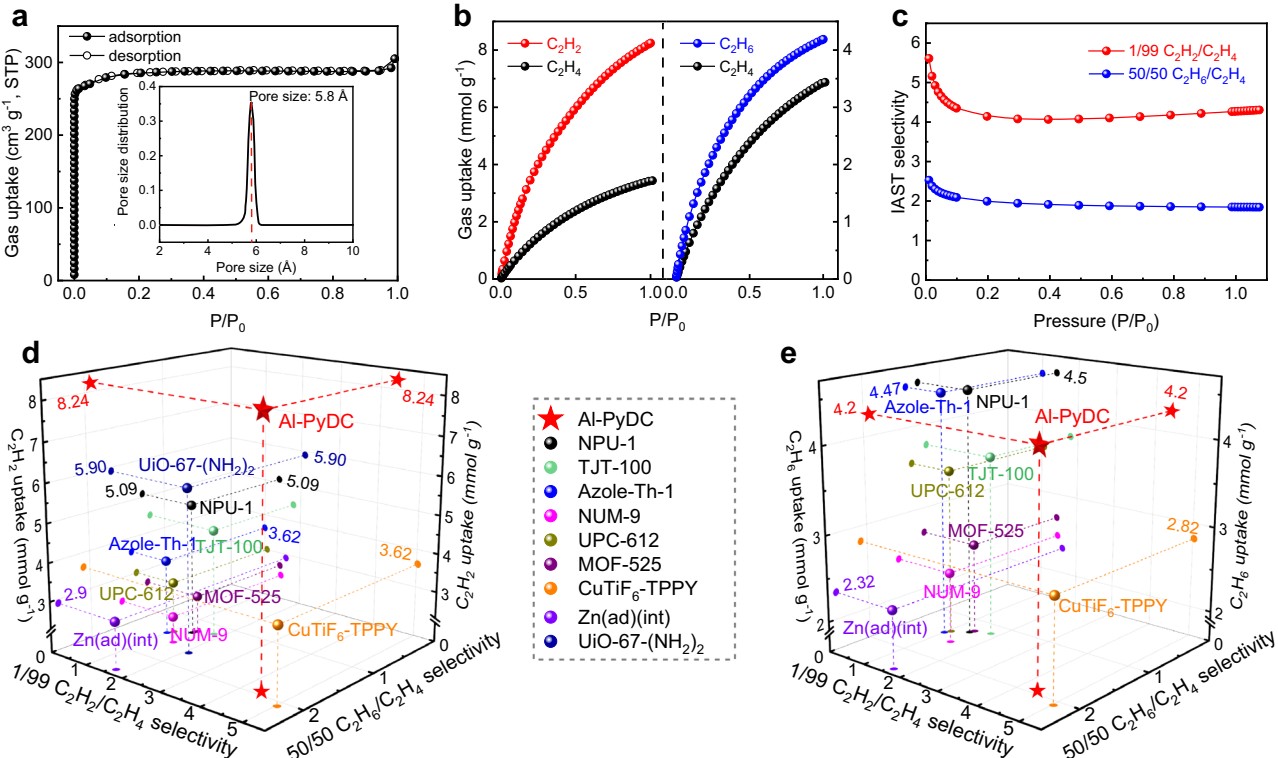

**Fig. 3 | Gas sorption properties. a** N$_2$ sorption isotherms of Al-PyDC at 77 K. Inset shows pore size distribution of Al-PyDC calculated based on NLDFT model. **b** Adsorption isotherms of Al-PyDC for C$_2$H$_2$, C$_2$H$_6$ and C$_2$H$_4$ at 296 K. **c** Predicted IAST selectivity curves for C$_2$H$_2$/C$_2$H$_4$ and C$_2$H$_6$/C$_2$H$_4$ mixtures at 296 K.

**d** Comparison of C$_2$H$_2$ uptake and gas selectivities for Al-PyDC with the promising C$_2$H$_2$/C$_2$H$_6$-selective materials at ambient conditions. **e** Comparison of C$_2$H$_6$ uptake and gas selectivities for Al-PyDC with the indicated materials.

investigated the time-dependent ads/desorption kinetics profiles of Al-PyDC at 296 K. As shown in Supplementary Fig. 7, Al-PyDC exhibits fast adsorption kinetics for all the C$_2$ molecules and the adsorbed molecules can be completely removed under a high vacuum quickly, mainly attributed to its comparably large pore channels.

These adsorption discrepancies between C$_2$ hydrocarbons can be partially explained by the experimental isosteric heat of adsorption ($Q_{st}$), calculated by adsorption isotherms at different temperatures (Supplementary Figs. 9–11). As shown in Supplementary Fig. 12, the calculated $Q_{st}$ values at zero loading were determined to follow the order of C$_2$H$_2$ (35.3 kJ mol$^{-1}$) > C$_2$H$_6$ (30.1 kJ mol$^{-1}$) > C$_2$H$_4$ (27.8 kJ mol$^{-1}$). Such higher $Q_{st}$ values for C$_2$H$_2$ and C$_2$H$_6$ compared with C$_2$H$_4$ confirm the stronger binding affinities of Al-PyDC with the former two gases, which is consistent well with that found in C$_2$ gas uptakes. Compared to most reported C$_2$H$_2$/C$_2$H$_6$-selective materials, Al-PyDC also shows relatively higher $Q_{st}$ values for C$_2$H$_2$ and C$_2$H$_6$ (Supplementary Fig. 13), probably attributed to the multiple supramolecular binding sites observed in Al-PyDC.

Ideal Adsorbed Solution Theory (IAST) was utilized to estimate the adsorptive selectivity of Al-PyDC for both 1/99 C$_2$H$_2$/C$_2$H$_4$ and 50/50 C$_2$H$_6$/C$_2$H$_4$ mixtures. As shown in Fig. 3c, Al-PyDC exhibits both high C$_2$H$_2$/C$_2$H$_4$ selectivity of 4.3 and C$_2$H$_6$/C$_2$H$_4$ selectivity of 1.9 at 296 K and 1 bar. The C$_2$H$_2$/C$_2$H$_4$ selectivity is the highest among all of the reported C$_2$H$_2$/C$_2$H$_6$-selective MOFs except CuTiF$_6$-TPPY[53], significantly larger than that of some best-performing materials including Zn(ad)(int) (1.61)[54], TJT-100 (1.8)[55], and NPU-1 (1.4)[56] and UiO-67-NH$_2$ (2.1)[59]. Furthermore, the C$_2$H$_6$/C$_2$H$_4$ selectivity of Al-PyDC is also among the highest for the existing C$_2$H$_2$/C$_2$H$_6$-selective MOFs, outperforming Azole-Th-1 (1.46)[49], TJT-100 (1.8)[55], UiO-67-NH$_2$ (1.7)[59] and other benchmark materials. As shown in Figs. 3d, e, when we set the uptakes and selectivities as concurrent objectives for both C$_2$H$_2$/C$_2$H$_4$ and C$_2$H$_6$/C$_2$H$_4$ separations, Al-PyDC exhibits by far the best

combination of very high uptakes and selectivities toward both separations, making it as the current benchmark for one-step separation of C$_2$H$_4$ from ternary C$_2$ mixtures.

## Binding-site determination

To visualize the locations of all C$_2$ molecules and thus elucidate the origin of the observed higher C$_2$H$_6$ and C$_2$H$_2$ uptakes over C$_2$H$_4$, we carried out the in situ SCXRD experiments on gas-loaded Al-PyDC crystals. The SCXRD data were collected at 200 K and 1 bar. According to the C$_2$H$_2$-loaded SCXRD analysis, Al-PyDC was found to exhibit two types of binding sites (site-I and site-II) for the adsorbed C$_2$H$_2$ molecules (Fig. 4a). As shown in Fig. 4b, the adsorbed C$_2$H$_2$ molecule in site-I is mainly located at the corner of square-shaped pore channel. This C$_2$H$_2$ molecule exhibits multiple interactions with six carboxylate oxygen atoms through six C−H⋯O hydrogen bonds with the distances of 2.47–3.30 Å. In addition, C$_2$H$_2$ molecule also interacts with μ-OH site of the Al-octahedral chain through two supramolecular interactions (O−H⋯C = 1.98 Å), and with two C−H groups from surrounding pyrrole rings through weak supramolecular interactions (C$_{C2H2}$⋯H$_{pyrrole}$ = 2.40 Å). The C$_2$H$_2$ binding site-II was found to be located around the channel wall and in close proximity to the PyDC linker (Fig. 4c). The site-II acetylene molecule interacts with three carboxylate oxygen atoms through C−H⋯O hydrogen bonds (2.41–3.35 Å) and with one pyrrole ring to form the van der Waals (vdW) interaction (C−H⋯π = 3.18 Å). Further, the C$_2$H$_2$ molecule also binds with two N−H groups from two PyDC linkers through weak supramolecular interactions (N−H⋯C$_{C2H2}$ = 2.32/2.88 Å). Such multiple supramolecular interactions in site-I and site-II were found to cooperatively interact with the adsorbed C$_2$H$_2$ molecules. Most importantly, as shown in Fig. 4d and Supplementary Fig. 14, due to the dense distribution of these two types of binding sites within the channels, each adsorbed C$_2$H$_2$ molecule in site-I interacts with two adjacent C$_2$H$_2$

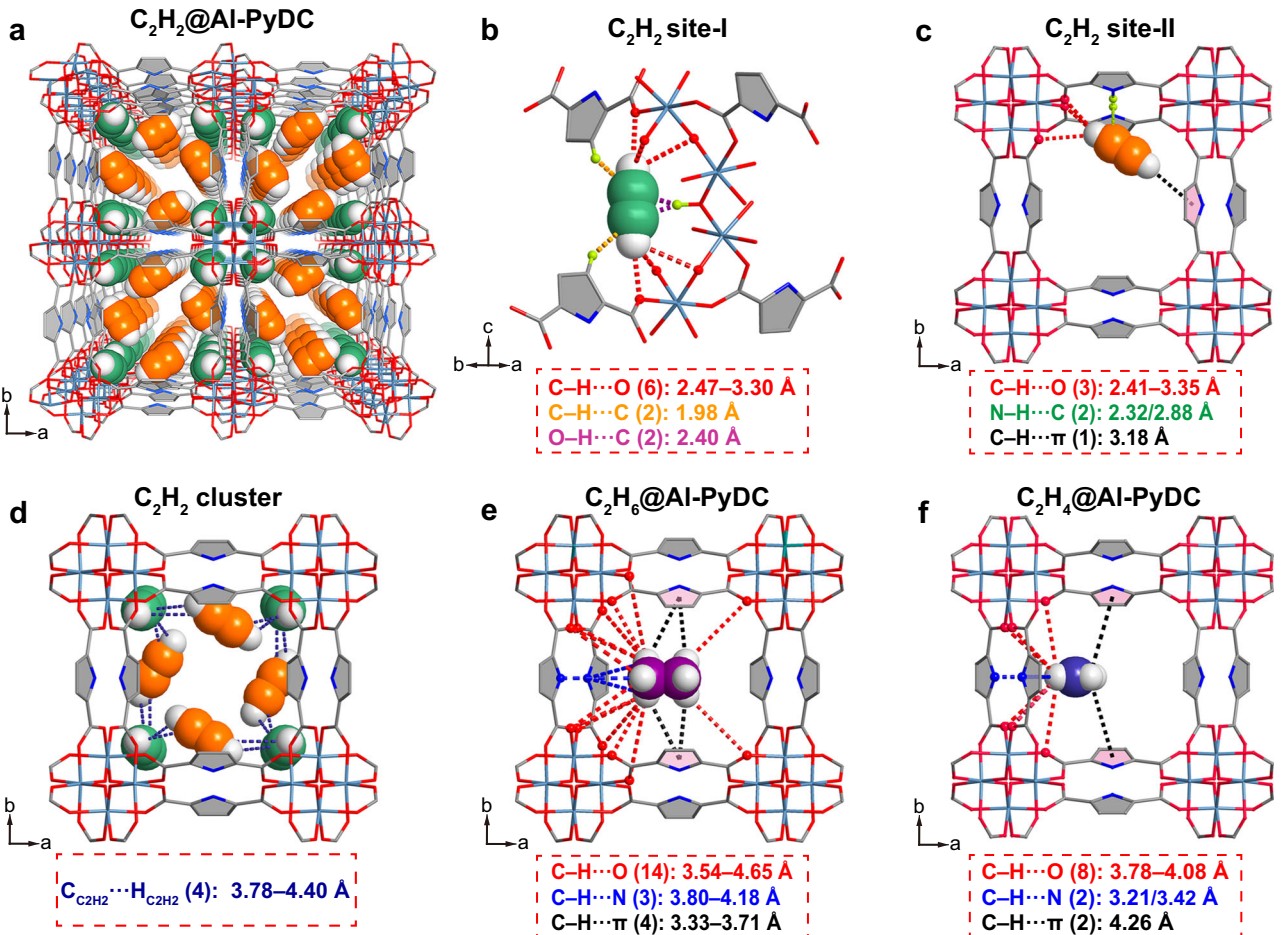

**Fig. 4 | Adsorption binding sites of $C_2H_2$, $C_2H_6$ and $C_2H_4$ in Al-PyDC. a** The SCXRD structure of $C_2H_2$-loaded Al-PyDC viewed along the *c* axis, indicating two types of $C_2H_2$ binding sites. **b** Illustration of $C_2H_2$ binding site-I, (**c**) $C_2H_2$ binding site-II, and (**d**) dense packing of $C_2H_2$ molecules within Al-PyDC. **e** The $C_2H_6$ binding site and (**f**) $C_2H_4$ binding site in gas-loaded Al-PyDC, determined by SCXRD analysis.

molecules in site-II through four C−H⋯C−H interactions (3.78–4.40 Å). Such cooperative interactions between guest molecules enable the dense packing of $C_2H_2$ molecules inside the pore channels, resulting in the ultrahigh $C_2H_2$ uptake capacity of Al-PyDC. Full occupation of these adsorption sites corresponds to 10.1 mmol g$^{-1}$ gas uptake, which is close to the saturated $C_2H_2$ uptake (11.1 mmol g$^{-1}$) at 196 K (Supplementary Fig. 15). This also implies that about 81.6% of these $C_2H_2$ adsorption sites are occupied at 296 K and 1 bar.

For $C_2H_6$ molecule, the adsorption location is approximately the same as that of $C_2H_2$ binding site-II. Due to its larger molecular size, $C_2H_6$ adopts an adsorption orientation different from that of site-II $C_2H_2$ molecule because of the steric restriction. The $C_2H_6$ orientation is perpendicular to the PyDC linker axis. As shown in Fig. 4e, multiple supramolecular interactions were observed between the adsorbed $C_2H_6$ molecule and the host framework. Each $C_2H_6$ molecule is H-bonded to twelve carboxylate O atoms around the pore through fourteen C−H⋯O interactions with the distances of 3.54–4.65 Å. In addition, $C_2H_6$ molecule also interacts with two pyrrole N atoms through three C−H⋯N interactions (3.80–4.18 Å) and with pyrrole ring on both sides through four C−H⋯π interactions (3.33–3.71 Å). In comparison, the adsorbed $C_2H_4$ molecules are located at the similar positions in the square-shaped channels. As depicted in Fig. 4f, each $C_2H_4$ molecule interacts with carboxylate O atoms to form only six C−H⋯O H-bonds with the distances of 3.83–4.08 Å, with two pyrrole N atoms through C−H⋯N interactions (3.21 and 3.42 Å) and with pyrrole ring on both sides through two C−H⋯π interactions (4.26 Å). Evidently, due to the more number of H

atoms and larger molecular size of $C_2H_6$, the adsorbed $C_2H_6$ molecule exhibits much more number of weak supramolecular interactions with the framework, leading to the stronger binding affinity than $C_2H_4$ molecule. This can be further confirmed by the higher experimental binding energies of $C_2H_6$ (30.1 kJ mol$^{-1}$) than $C_2H_4$ (27.8 kJ mol$^{-1}$) observed in Al-PyDC. All of the above results can visually elucidate the adsorption and separation phenomenon on $C_2$ gas mixtures.

## Dynamic breakthrough studies

Dynamic breakthrough experiments were first conducted on a packed column filled with activated Al-PyDC to evaluate the separation performance for binary $C_2H_6/C_2H_4$ (50/50) and $C_2H_2/C_2H_4$ (1/99) mixtures under a gas flow of 1.25 mL min$^{-1}$ at 296 K. As shown in Fig. 5a, owing to the ultrahigh $C_2H_2$ uptake and large $C_2H_2/C_2H_4$ selectivity, Al-PyDC exhibits a highly efficient separation capacity for 1/99 $C_2H_2/C_2H_4$ mixture, wherein $C_2H_4$ gas first eluted through the adsorption bed at 34 min, while $C_2H_2$ breakthrough did not occur until 92 min. During this time interval, pure $C_2H_4$ production (> 99.999%) from the outlet effluent for a given cycle was calculated to be 7.93 mmol g$^{-1}$. This $C_2H_4$ productivity is even much higher than some top-performing materials for single $C_2H_2/C_2H_4$ separation, such as UTSA-100a (1.13 mmol g$^{-1}$)[11] and SIFSIX-3-Zn (1.94 mmol g$^{-1}$)[37]. Further, the efficient separation of $C_2H_4$ from $C_2H_6/C_2H_4$ mixture can be also accomplished by Al-PyDC (Fig. 5b), with a high pure $C_2H_4$ productivity of 0.68 mmol g$^{-1}$ at the outlet. This productivity is higher than some promising $C_2H_6$-selective materials such as Cu(Qc)$_2$ (0.42 mmol g$^{-1}$)[42], MAF-49

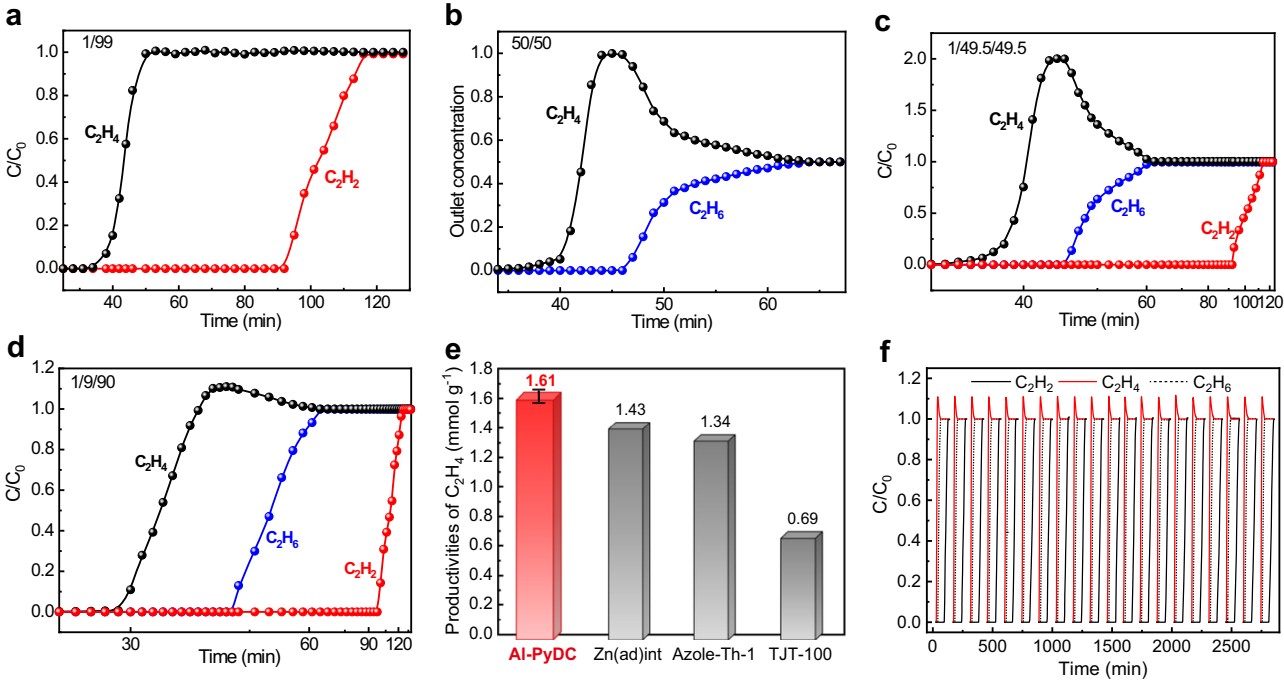

**Fig. 5 | Dynamic breakthrough experiments.** Experimental column breakthrough curves of Al-PyDC for (**a**) $C_2H_2/C_2H_4$ (1/99), (**b**) $C_2H_6/C_2H_4$ (50/50) and (**c**) $C_2H_2/C_2H_6/C_2H_4$ (1/49.5/49.5) mixtures under ambient conditions. **d** Experimental column breakthrough curves of Al-PyDC for $C_2H_2/C_2H_6/C_2H_4$ (1/9/90) mixtures under ambient conditions. **e** Comparison of pure $C_2H_4$ productivity for Al-PyDC (the averaged value was obtained from five independent tests, and the error bar is the standard deviation) with the top-performing materials reported. **f** Twenty separation cycles of breakthrough experiments on Al-PyDC for $C_2H_2/C_2H_6/C_2H_4$ (1/9/90) mixture.

(0.28 mmol g⁻¹)[65] and even comparable to the benchmark $Fe_2(O_2)$(dobdc)₂ (0.79 mmol g⁻¹)[14].

Next, we examined the separation performance of Al-PyDC for actual $C_2H_2/C_2H_6/C_2H_4$ ternary mixtures at the same conditions. Figure 5c reveals that complete separation of both $C_2H_6$ and $C_2H_2$ from 3-component $C_2H_2/C_2H_6/C_2H_4$ (1/49.5/49.5) mixture can be fulfilled by Al-PyDC. It was found that $C_2H_4$ was first eluted at 33 min to yield a pure gas with an undetectable amount of $C_2H_6$ and $C_2H_2$ (the detection limit of the instrument is 0.01%). After that, $C_2H_6$ gas secondly passed through the adsorption bed at 46 min, and the adsorbent retained $C_2H_2$ lastly until 92 min. During the time interval between $C_2H_4$ and $C_2H_6/C_2H_2$ breakthrough, pure $C_2H_4$ productivity of Al-PyDC from the outlet effluent for a given cycle was calculated up to 0.66 mmol g⁻¹. This $C_2H_4$ productivity is even higher than the previously benchmark UiO-67-(NH₂)₂ (0.55 mmol g⁻¹)[59]. Given the fact that the $C_2$ fraction obtained from cracked gas sometimes contains a small portion of $C_2H_6$ (ca. 6–10%), we further evaluated its separation capacity on a ternary $C_2H_2/C_2H_6/C_2H_4$ (1/9/90) mixture at the same conditions. As shown in Fig. 5d, the much later breakthrough times of both $C_2H_2$ and $C_2H_6$ (100 and 43 min) than that of $C_2H_4$ (29 min) reveal that simultaneous removal of $C_2H_2$ and $C_2H_6$ can be fulfilled by Al-PyDC. The productivity of pure $C_2H_4$ (over 99.9%) from the outlet effluent was calculated up to 1.61 mmol g⁻¹ (Fig. 5e), which is the highest among the reported best-performing $C_2H_2/C_2H_6$-selective materials including Azole-Th-1 (1.34 mmol g⁻¹, 1/9/90)[49], Zn(ad)(int) (1.43 mmol g⁻¹, 1/10/89)[54] and TJT-100 (0.69 mmol g⁻¹, 0.5/0.5/99)[55]. The highly efficient separation capacity can be still retained when the flow rate of the mixed gases accelerated to 5.0 and 10.0 mL min⁻¹ (Supplementary Fig. 17).

For practical applications, the adsorbent should possess good recyclability and structural stability. To test the recyclability of Al-PyDC, we firstly carried out cycling experiments on Al-PyDC for single-component $C_2H_2$ or $C_2H_6$ adsorption at 1 bar and 296 K, followed by desorption under vacuum at room temperature. As shown in Supplementary Fig. 18, the experimental cycling results indicate that there was no noticeable loss in $C_2H_2$ and $C_2H_6$ adsorption capacities for Al-PyDC over 30 cycles. Next, breakthrough separation experiments on both $C_2H_2/C_2H_6/C_2H_4$ (1/49.5/49.5) and $C_2H_2/C_2H_6/C_2H_4$ (1/9/90) mixtures were cycled numerous times to further assess the recyclability of Al-PyDC (Fig. 5f and Supplementary Figs. 19–21), wherein the sample was fully regenerated between cycles under sweeping He gas at 373 K. The cycling results show that there was no obvious loss in the separation capacity for Al-PyDC over 20 cycles, confirming its good recyclability for this separation. Given that the actual feed streams typically contain small amount of acidic gases (e.g., $H_2S$),[67] the breakthrough experiments for 1/9/90 $C_2H_2/C_2H_6/C_2H_4$ mixture with 1000 ppm $H_2S$ were performed on Al-PyDC to investigate the effect of acidic $H_2S$ gas on separation performance. As shown in Supplementary Fig. 22, the separation performance of Al-PyDC remains almost unchanged to afford the comparable $C_2H_4$ productivity of 1.57 mmol g⁻¹, and the separation capacity can be preserved over six continuous cycles under the presence of acid gases. These results have demonstrated that Al-PyDC can be recycled for repeated separation cycles without any loss of performance even under the presence of acid gases.

## Material stability and scale-up synthesis

Besides separation performance, material stability and scale-up synthesis are two most important concerns for industrial applications. We first examined the chemical stability of Al-PyDC after treatment under different conditions, monitored by powder X-ray diffraction (PXRD) and gas adsorption measurements. As shown in Fig. 6a, after immersion in water, boiling water, and aqueous solutions of pH 1 and 12 for 3 days, the PXRD studies revealed that the framework of Al-PyDC can retain its structural integrity without any phase change and loss of crystallinity observed. Such ultrahigh chemical stability was also confirmed by $N_2$ adsorption isotherms at 77 K and $C_2H_2/C_2H_6$ uptake capacities at 296 K after different treatment, wherein all the gas adsorption amounts show no obvious

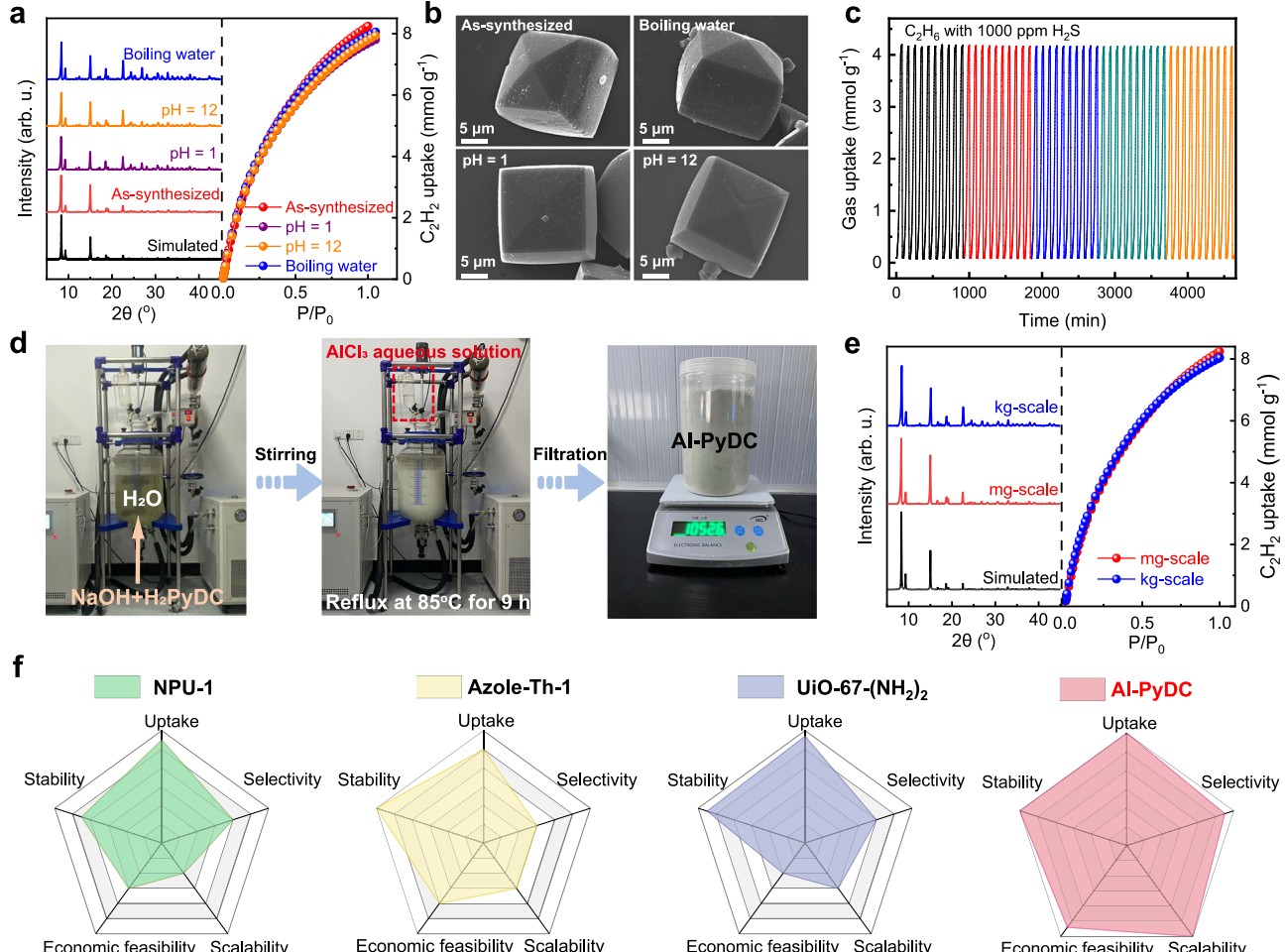

**Fig. 6 | Stability and scalable synthesis. a** PXRD patterns and $C_2H_2$ adsorption isotherms and (**b**) SEM images of Al-PyDC samples after treatment with different conditions. **c** Gas adsorption on Al-PyDC over 50 consecutive adsorption-desorption cycles for $C_2H_6$ gas containing 1000 ppm $H_2S$ at 296 K, in which the sample was regenerated by $N_2$ sweeping at 373 K. **d** The kilogram-scale synthesis of Al-PyDC by a green and facile method. **e** PXRD patterns and $C_2H_2$ adsorption isotherms of Al-PyDC samples obtained from various-scale synthesis. **f** The comprehensive comparison of separation performance, stability, economic feasibility and scalability for Al-PyDC and other benchmark materials.

decrease compared to those of the pristine sample (Supplementary Fig. 23). In addition, scanning electron microscope (SEM) and optical images of Al-PyDC crystals showed that there are no obvious changes observed in their morphology and surface after the treatment with water and pH solutions (Fig. 6b and Supplementary Fig. 24). The variable temperature PXRD patterns indicate that Al-PyDC is thermally stable up to 350 °C (Supplementary Fig. 25). Therefore, this material shows one of the best chemical and thermal stabilities among the reported MOFs (Supplementary Table 8), even comparable to some representative stable MOFs including MIL-101(Cr), BUT-12 and PCN-250.[68–70] Considering that the actual working environment would contain trace amount of acid gases in feed streams[67], we further assess the stability of Al-PyDC under acid gas circumstances. We first carried out the repeated adsorption/desorption cycling experiments for $C_2H_6$ gas containing 1000 ppm acidic $H_2S$ gas, in which the sample was regenerated by $N_2$ sweeping at 373 K. As shown in Fig. 6c, the $C_2H_6$ adsorption capacity can be maintained with no noticeable loss after 50 cycles in the presence of acidic $H_2S$. Further, the breakthrough cycling experiments on $C_2H_2/C_2H_6/C_2H_4$ mixtures containing 1000 ppm $H_2S$ further confirmed that the separation capacity of Al-PyDC can be preserved without any loss of performance over six cycles under the presence of acid gases (Supplementary Fig. 22). The PXRD pattern and 77 K $N_2$ sorption isotherms after repeated breakthrough cycles indicate

that Al-PyDC can maintain its structural integrity (Supplementary Fig. 26). These results confirm its high structural stability under practical circumstances.

With respect to large-scale practical applications, it is crucial that the scale-up synthesis of MOFs should be viable from a technical and economic point of view. The cost analysis for various MOF syntheses indicates that the raw material costs are often prohibitively high, especially for organic linkers or using organic solvents as reaction solutions. Further, synthetic conditions can also have an important influence on the economics. For instance, the necessary use of high-temperature or high-pressure apparatus for MOF syntheses would be not only costly but also bring high expenses on safety precautions. Al-PyDC exhibits none of these disadvantageous conditions. The reported literature showed that Al-PyDC can be prepared from simple and commercially available reagents of $H_2PyDC$ and $Al_2(SO_4)_3 \cdot 18H_2O$ in water solvent at a temperature of 120 °C[65]. However, the synthetic temperature (120 °C) is still much higher than the water boiling point, which is detrimental to the scale-up synthesis. To reduce the synthetic temperature, we here optimized the reaction conditions by changing the base equivalents and aluminum salt (see the Method for details). Our experiments revealed that Al-PyDC can be readily synthesized from $H_2PyDC$ and $AlCl_3 \cdot 6H_2O$ in water solvent at a much lower temperature (85 °C). These mild and water-based conditions can be considered as a scalable green synthesis, which are particularly

advantageous from safety and environmental aspects. As shown in Fig. 6d and Supplementary Fig. 27, the kilogram-scale production of Al-PyDC sample was easily performed by using a reflux-based synthesis method in a 30 L reaction vessel. This protocol provided more than 1.0 kg of as-synthesized Al-PyDC per reaction batch in a high yield of 92% within 9 h, resulting in an exceptional space-time yield (STY) over 126 kg m$^{-3}$ day$^{-1}$. For a comparison, the STYs of zeolites are commonly in the range of 50–150 kg m$^{-3}$ day$^{-1}$[12]. The Al-PyDC products synthesized at this large-scale exhibit similar crystallinity and gas uptake capacities compared to the material produced at a small scale, as verified by PXRD analysis and gas adsorption isotherms (Fig. 6e and Supplementary Fig. 29). Further, dynamic breakthrough experiments on activated Al-PyDC sample synthesized in this low temperature procedure showed the same separation capacity for ternary C$_2$ hydrocarbon mixtures (Supplementary Fig. 29), without any loss in C$_2$H$_4$ productivity.

A perfect adsorbent for industrial one-step C$_2$H$_4$ purification from ternary mixtures should meet the following criteria: (1) large C$_2$H$_2$ and C$_2$H$_6$ adsorption capacities; (2) high C$_2$H$_2$/C$_2$H$_4$ and C$_2$H$_6$/C$_2$H$_4$ selectivities; (3) viable chemical/thermal stability; (4) economic feasibility; (5) easy scalability of production. As shown in Fig. 6f, we comprehensively compare all the above criteria between Al-PyDC and other benchmark materials. We have shown that Al-PyDC can meet all of these criteria. In comparison, other benchmark MOFs have better reported properties in one or more of the above-mentioned criteria, but not in all of them. For instance, several benchmark MOFs (e.g., Azole-Th-1, NPU-1, and UiO-67-NH$_2$) show highly efficient separation performance for one-step C$_2$H$_4$ purification; however, they suffer from either the use of toxic organic solvents (e.g., dimethylformamide) and harsh synthesis conditions, or contain expensive and complicated organic linkers (Supplementary Table 8 and Supplementary Fig. 30). These drawbacks lead to an extremely high difficulty on the scale-up production of these materials, making most of them unfavorable for large-scale industrial applications. By far, there are no reports existed on kilogram-scale synthesis of MOFs relevant for one-step C$_2$H$_2$/C$_2$H$_6$/C$_2$H$_4$ separation. With Al-PyDC, we demonstrated that a high-yielding (>90%) and scalable (~1.05 kg) synthesis from simple and commercially available reagents was achieved in water solution and with a high STY value, making the cost of Al-PyDC synthesis more affordable. In terms of separation performance, Al-PyDC exhibits one of the highest C$_2$H$_2$ and C$_2$H$_6$ uptakes and selectivities over C$_2$H$_4$ at ambient conditions, providing a maximum pure C$_2$H$_4$ productivity for ternary mixtures. Overall, the combined superiorities of green synthesis method, benchmark separation performance, high stability, economic feasibility and easy scalability of production make this material as a promising adsorbent to address the challenges for industrial one-step C$_2$H$_4$ purification from ternary mixtures.

## Discussion

In summary, we have proposed and demonstrated a scalable and robust Al-MOF (Al-PyDC) with multiple supramolecular binding sites for highly efficient one-step C$_2$H$_4$ purification from ternary C$_2$ mixtures. The gas-loaded SCXRD studies of Al-PyDC visually identified that the low-polarity pore surfaces with abundant O/N sites provide stronger multiple supramolecular interactions with C$_2$H$_2$ and C$_2$H$_6$ over C$_2$H$_4$, thus simultaneously optimizing the C$_2$H$_2$ and C$_2$H$_6$ adsorption uptakes and selectivities. This material thereby achieves the top-tier C$_2$H$_2$ and C$_2$H$_6$ uptakes (8.24 and 4.20 mmol g$^{-1}$) and selectivities (4.3 and 1.9) over C$_2$H$_4$ at ambient conditions, outperforming most of the benchmark materials reported to date. The breakthrough experiments affirmed that Al-PyDC can simultaneously separate C$_2$H$_2$ and C$_2$H$_6$ from ternary C$_2$H$_2$/C$_2$H$_6$/C$_2$H$_4$ mixtures, affording the record high polymer-grade C$_2$H$_4$ productivity of 1.61 mmol g$^{-1}$. Most remarkably, Al-PyDC is highly water/pH stable and can be easily produced at kilogram-scale using a green synthesis

method. The comprehensive features of high separation performance, notable stability/recyclability, economic feasibility and easy scalability of synthesis make this material as the current benchmark for industrial one-step C$_2$H$_4$ purification applications. This work also provides an effective approach of designing multiple supramolecular binding sites in microporous MOFs to concurrently enforce C$_2$H$_2$ and C$_2$H$_6$ binding affinity for boosting this important gas separation.

## Methods

### Materials

Aluminum chloride hexahydrate (AlCl$_3$·6H$_2$O, CAS: 231-208-1, purity 99.9%) was purchased from Aladdin, 1H-Pyrrole-2,5-dicarboxylic acid (H$_2$PyDC, CAS: 937-27-9, purity 97%) was purchased from Bidepharm (China). N$_2$ (99.999%), C$_2$H$_4$ (99.9%), C$_2$H$_6$ (99.99%), C$_2$H$_2$ (99.6%), He (99.999%) and mixed gases of C$_2$H$_2$/C$_2$H$_4$ (1/99, v/v), C$_2$H$_6$/C$_2$H$_4$ (50/50), C$_2$H$_2$/C$_2$H$_6$/C$_2$H$_4$ (1/49.5/49.5) and C$_2$H$_2$/C$_2$H$_6$/C$_2$H$_4$ (1/9/90) were purchased from JinGong Company (China). All chemicals were used without further purification.

### Synthesis of Al-PyDC powder

The Al-PyDC powder sample was synthesized based on the method reported in the previous literature with modification[65]. H$_2$PyDC (310 mg, 2 mmol) and NaOH (240 mg, 6 mmol) were dissolved in H$_2$O (6 mL) by sonication for 10 min to obtain clear solution. Afterwards, aqueous AlCl$_3$ solution (1 M, 2 mL) was added. The reaction mixture was then kept at 85 °C under refluxed condition for 9 h. The resulting white precipitate of Al-PyDC was separated by using filtration and washed with H$_2$O and ethanol.

### Crystallization of Al-PyDC

Single crystals were obtained by fully dissolving H$_2$PyDC (39 mg, 0.25 mmol) in aqueous NaOH solution (0.05 M, 5 mL). Afterwards, aqueous AlCl$_3$ solution (0.05 M, 5 mL) was slowly added. The resulting solution was incubated for 48 h in a pre-heated oven at 100 °C.

### Scalable synthesis of Al-PyDC

H$_2$PyDC (0.9 kg, 5.7 mol) and NaOH (0.684 kg, 17.1 mol) were dissolved in deionized water (17 L) at room temperature with stirring for 30 min to obtain clear solution. Afterwards, AlCl$_3$·6H$_2$O (1.374 kg, 5.7 mol) was dissolved in 3 L deionized water and transferred to a glass material-feeding funnel. The AlCl$_3$ aqueous solution was added at a rate of 3 L per hour to the reaction vessel, resulting in the formation of white precipitate. The solution was then heated and kept at 85 °C for 9 h under reflux condition with the spinner rotating at 150 rpm. After that, the solid product was collected in a 20 L filtration funnel and thoroughly washed with deionized water and ethanol, then vacuum dried at 393 K overnight to obtain ~1.05 kg product with a yield of 92%.

### Sample characterization

The SEM images were obtained from a Hitachi S4800 field-emission scanning electron microscopy (FE-SEM). The microscopy images were captured on an Olympus IX 71 inverted fluorescent microscope. The PXRD patterns in the 2θ = 2–45° range were measured on an X'Pert PRO diffractometer at room temperature using a Cu-Kα ($\lambda$ = 1.54184 Å) radiation source. Thermogravimetric analysis (TGA) was measured on a TA SDT-650 instrument and the sample was heated under N$_2$ flow (50 mL min$^{-1}$) with a heating rate of 5 K min$^{-1}$.

### Gas sorption measurements

Before the test, the as-synthesized sample was solvent-exchanged with high-purity methanol over eight times within 3 days to the complete removal of guest solvent molecules from the framework. The solvent-exchanged sample was degassed for 12 h at room temperature and then for another 12 h at 393 K until the outgas rate was 5 μmHg min$^{-1}$. Gas sorption isotherms of C$_2$H$_2$, C$_2$H$_6$, and C$_2$H$_4$ were obtained from a

Micromeritics ASAP 2020 instrument, and the adsorption tube was kept at a constant temperature of 273 K, 296 K and 313 K by using a Julabo water bath. $N_2$ sorption isotherms were recorded by using a Micromeritics ASAP 2460 instrument at 77 K under liquid $N_2$ bath. Kinetic and equilibrium adsorption measurements were carried out using an Intelligent Gravimetric Analyzer (IGA001, Hiden, UK) under diverse test conditions.

### Single-crystal X-ray diffraction
SCXRD data were collected at 170 K for Al-PyDC-hydrated, and at 200 K for $C_2H_2$@Al-PyDC, $C_2H_6$@Al-PyDC and $C_2H_4$@Al-PyDC on an Agilent Supernova CCD diffractometer equipped with graphite-monochromatic enhanced Cu-Kα radiation ($\lambda = 1.54184$ Å). A single crystal of solvent-exchanged Al-PyDC was selected and placed into a capillary glass tube with inner diameter of 0.1 mm. This crystal was activated at 393 K for 4 h, and the capillary glass tube was filled by pure $C_2H_2$, $C_2H_6$ or $C_2H_4$ gas up to 1 bar and then sealed to obtain $C_2H_2$-loaded, $C_2H_6$-loaded or $C_2H_4$-loaded Al-PyDC crystal. The datasets were corrected by empirical absorption correction using spherical harmonics, implemented in the SCALE3 ABSPACK scaling algorithm. The structure was solved by direct methods and refined by full matrix least-squares methods with the SHELX-97 program package. During crystal structure analysis for $C_2H_6$@Al-PyDC, we found that guest $C_2H_6$ molecules exhibited highly positional disorder within the channels. By atomic identification and refinement, we determined that the asymmetric unit of each $C_2H_6$ molecule within the channel includes four carbon atoms (C5, C6, C8, C9) and eight hydrogen atoms (H5A, H5C, H6A, H6C, H8A, H8C, H9A, H9C). Among them, four carbon atoms and four hydrogen atoms (H5A, H6A, H8A, H9A) are located on a mirror plane ($-x, y, z$), while two carbon atoms (C6 and C8) are located on a mirror plane ($x, 1/2-y, z$) and a 2-fold rotation axis ($1/2-x, y, 1/2-z$). The asymmetric unit of $C_2H_6$ is generated by these symmetry operations, resulting in four completely disordered $C_2H_6$ molecules at each site, with an occupancy of 25%. All the crystal data are summarized in Supplementary Table 4, and ORTEP style illustrations of all structures are provided in Supplementary Figs. 31–34.

### Breakthrough experiments
The breakthrough curves were obtained from a dynamic gas breakthrough apparatus equipped with stainless steel column ($\Phi$ 4 × 120 mm). The weight of sample packed in the column was 0.34 g. The mixed gas flows of (1) $C_2H_2$/$C_2H_4$ (1/99, v/v), (2) $C_2H_6$/$C_2H_4$ (50/50), (3) $C_2H_2$/$C_2H_6$/$C_2H_4$ (1/49.5/49.5) or (4) $C_2H_2$/$C_2H_6$/$C_2H_4$ (1/9/90) were introduced into breakthrough apparatus with the rate of 1.25, 5 and 10 mL min$^{-1}$ at 296 K and 1 bar, respectively. The outlet gas was monitored using a gas chromatography (GC-2014C, SHIMADZU) equipped with thermal conductivity detector (TCD, detection limit 0.1 ppm). The concentration of the outlet gas was calibrated by detecting the standard gas mixture. After each breakthrough experiment, the sample can be regenerated under a purging He gas with a flow of 10 mL min$^{-1}$ at 373 K for 1 h.

### Cycling experiments
The cycling experiments for single-component $C_2H_2$ or $C_2H_6$ were performed on a Micromeritics ASAP 2020 surface area analyzer. The activated sample was exposed to $C_2H_2$ or $C_2H_6$ gas at 1 bar and 296 K until saturation, followed by desorption under vacuum at room temperature to 0.001 bar. The cycling experiments for $C_2H_6$ with 1000 ppm $H_2S$ were measured by using DSC/TGA Discovery SDT 650. After being fully activated at 393 K under $N_2$ flow for 2 h, the adsorbent was exposed to $C_2H_6$ gas containing 1000 ppm $H_2S$ (100 mL min$^{-1}$) at 296 K for 60 min, followed by regeneration at 373 K under $N_2$ flow (200 mL min$^{-1}$) for 30 min. Cycling breakthrough experiments for $C_2H_2$/$C_2H_6$/$C_2H_4$ mixture were performed using a dynamic gas breakthrough apparatus equipped with stainless steel column. The weight of

sample packed in the column was 0.418 g. The mixed gas flows of (1) $C_2H_2$/$C_2H_6$/$C_2H_4$ (1/49.5/49.5), (2) $C_2H_2$/$C_2H_6$/$C_2H_4$ (1/9/90) or (3) $C_2H_2$/$C_2H_6$/$C_2H_4$ (1/9/90) with 1000 ppm $H_2S$ were introduced into breakthrough apparatus with the rate of 1.25 mL min$^{-1}$ at 296 K and 1 bar. After each breakthrough experiment, the sample can be fully regenerated under a purging He gas with a flow of 10 mL min$^{-1}$ at 373 K for 30 min.

## Data availability
Crystallographic data for the structures in reported this article have been deposited at the Cambridge Crystallographic Data Centre, under deposition numbers CCDC 2242152−2242155. Copies of the data can be obtained free of charge via https://www.ccdc.cam.ac.uk/structures/. All the other relevant data that support the findings of this study are available within the article and its Supplementary Information, or from the corresponding author upon request.

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

## Acknowledgements

This research was supported by the National Science Foundation of China (52073251, B.L. and U22A20251, G.Q.), the Zhejiang Provincial Natural Science Foundation of China (No. LR22E030003, B.L.), and the Science Technology Department of Zhejiang Province (2022C01225, G.Q.).

## Author contributions

E.W. and X.-W.G. synthesized and characterized the MOF samples, and measured the adsorption isotherms. E.W., X.Z., H.W., and W.Z. collected and analyzed the single-crystal X-ray diffraction data. E.W., X.-W.G., and D.L. measured and analyzed the breakthrough data. B.L. and G.Q. directed and supervised the project. B.L. wrote the paper, and all authors contributed to revising the manuscript.

## Competing interests

The authors declare no competing interests.
