## [Peer Review File · Nature Communications]

REVIEWER COMMENTS

Reviewer #1 (Remarks to the Author):

Achieving efficient separation of ethylene from the hydrocarbon mixtures is a daunting challenge requiring appropriate tuning of the interactions between the host and the guest adsorbate molecules. Moreover, the development of such processes needs to be executed in a sustainable manner for a large-scale applicability. In this submission, the authors have utilized the interesting interplay of low polarity pore surfaces and densely distributed N/O binding sites providing multiple supramolecular interactions for preferential binding of acetylene and ethane over ethylene. Moreover, the authors have also demonstrated a green, scalable synthesis of the material Al-PyDC with excellent yield. The overall idea is quite intriguing, and significant improvements have been achieved over the existing materials in terms of selectivity from ternary mixtures. I can therefore recommend this work for publication in Nature Communications after the following revisions have been carried out.

1. Some more references relevant to this work need to be cited. For example, Nat Commun 12, 6507 (2021); ACS Appl. Mater. Interfaces 2019, 11, 34, 31499–31507; Eur. J. Inorg. Chem. 2021, 2021, 4498. Moreover, MOFs such as ATC-Cu (Angew. Chem. Int. Ed. 2021, 60, 5283-5288) and MIL-160 (J. Am. Chem. Soc. 2022, 144, 1681-1689) with the C₂H₂ uptake capacities of 5.01 mmol/g and 7.7 mmol/g need to be included as well.

2. A number of MOFs have been developed that show high chemical and thermal stabilities. In the "Material stability and scale-up synthesis" section on Page 18, the authors should cite and do a comparison with the reported stable MOFs (Adv. Mater. 2018, 30, 1704303; Chem. Soc. Rev., 2022, 51, 6417-6441).

3. The authors mention that "Al-PyDC exhibits an obviously preferential adsorption of C₂H₂ and C₂H₆ over C₂H₄ at all the temperatures". However, at 273 K, the C₂H₄ and C₂H₆ uptakes are quite similar, with the C₂H₄ uptake slightly higher at 1 bar. An explanation needs to be provided for this discrepancy, with a more detailed discussion on effect of temperature and the observed uptake capacities at 273 and 313 K.

4. Why does the Q_{st} for C₂H₆ show an initial increase with the uptake, followed by a decrease?

5. The disorder of C₂H₆ in the structure needs to be explained well. Also, the very short C6-C6 bond (1.09 Å) in C₂H₂ loaded structure needs to be checked.

6. The grammar needs improvement in some parts of the manuscript. For example:

Page 4, line 78: "the weakness nature" should be "the weak nature"

Page 4, lines 82-84: "UiO-67-(NH₂)₂ or CuTiF₆-TPPY holds the benchmark C₂H₂ and C₂H₆ uptakes or selectivities, but limited by inadequate gas selectivities or uptakes" is a bit confusing. Replacing "or" with "and" will make the sentence more clear.

Page 4, line 91: "no reports existed" should be "no reports existing".

Page 7, line 148: "octahedrally AlO₆-polyhedras" should be "octahedral AlO₆-polyhedras"

Page 10, line 192: "It is worthy of note" should be "It is worthy to note".

Reviewer #2 (Remarks to the Author):

This study reported that MOF-313 has promising properties for separating C₂H₄ from ternary mixtures of C₂H₂, C₂H₄, and C₂H₆. The scientific innovation and technological significance were found in engineering elements such as strong hydrocarbon absorption and corresponding hydrocarbon selectivity. They also exhibited chemical stability and ease of manufacture of the materials. On the other hand, previous studies have already reported MOF-313 production techniques and adsorption-based separation kinetics of C₂H₄ from mixtures. Thus, in my opinion, the overall novelty of this work is weak, and so more engineering data should be provided for publication in this journal. Here are some specific comments.

1. A breakthrough experiment would be a useful indicator of adsorption-based mixture separation. It can, however, only describe one cycle of the adsorption process. This material should be evaluated in

repeated adsorption and desorption environments, and related engineering data should be provided to improve technological significance of this work.

2. Low temperature synthesis is one method of lowering production costs. It may, however, be associated with crystallinity deterioration and loss of selective function. The crystals formed in the low temperature procedure need to be tested for hydrocarbon separation.

3. I don't understand why aqueous environment was employed to confirm stability of the crystal. I think working environmental of the crystal toward the ternary mixture would be different, and may not include water vapors. It is necessary to assess the stability under actual or accelerated adsorption and desorption circumstances.

Reviewer 1

Comments: Achieving efficient separation of ethylene from the hydrocarbon mixtures is a daunting challenge requiring appropriate tuning of the interactions between the host and the guest adsorbate molecules. Moreover, the development of such processes needs to be executed in a sustainable manner for a large-scale applicability. In this submission, the authors have utilized the interesting interplay of low polarity pore surfaces and densely distributed N/O binding sites providing multiple supramolecular interactions for preferential binding of acetylene and ethane over ethylene. Moreover, the authors have also demonstrated a green, scalable synthesis of the material Al-PyDC with excellent yield. The overall idea is quite intriguing, and significant improvements have been achieved over the existing materials in terms of selectivity from ternary mixtures. I can therefore recommend this work for publication in Nature Communications after the following revisions have been carried out.

Response: Thank you very much for your very positive comments.

(1) Some more references relevant to this work need to be cited. For example, Nat Commun 12, 6507 (2021); ACS Appl. Mater. Interfaces 2019, 11, 34, 31499–31507; Eur. J. Inorg. Chem. 2021, 2021, 4498. Moreover, MOFs such as ATC-Cu (Angew. Chem. Int. Ed. 2021, 60, 5283-5288) and MIL-160 (J. Am. Chem. Soc. 2022, 144, 1681-1689) with the C₂H₂ uptake capacities of 5.01 mmol/g and 7.7 mmol/g need to be included as well.

Response: Thank you for your suggestion. We have included all the above references in refs 39-41, 58 and 61 in the revised manuscript accordingly, highlighted in yellow.

(2) A number of MOFs have been developed that show high chemical and thermal stabilities. In the “Material stability and scale-up synthesis” section on Page 18, the authors should cite and do a comparison with the reported stable MOFs (Adv. Mater. 2018, 30, 1704303; Chem. Soc. Rev., 2022,51, 6417-6441).

Response: Thank you very much for your suggestion. The above-mentioned references have been cited in refs. 68-70 in the revised manuscript accordingly. As suggested, we further made a comparison on the stability of Al-PyDC with some stable MOFs, such as MIL-101(Cr), UiO-66, BUT-12 and PCN-250. As shown in Supplementary Table 8 (Supplementary Information), the chemical stability of Al-PyDC is comparable to these representative stable MOFs. The related comments on stability are included in page 19 accordingly, highlighted in yellow.

(3) The authors mention that “Al-PyDC exhibits an obviously preferential adsorption of C₂H₂ and C₂H₆ over C₂H₄ at all the temperatures”. However, at 273 K, the C₂H₄ and C₂H₆ uptakes are quite similar, with the C₂H₄ uptake slightly higher at 1 bar. An explanation needs to be provided for this discrepancy, with a more detailed discussion on effect of temperature and the observed uptake capacities at 273 and 313 K.

Response: Thank you very much for pointing out this issue. Since the C_2H_6 binding affinity is stronger than C_2H_4 (Fig. R1 or Supplementary Fig. 12), it is normal that the C_2H_6 uptake is higher than C_2H_4 at different temperatures such as 296 K and 313 K. However, at 273 K, the C_2H_4 and C_2H_6 uptakes are quite similar at 1 bar, accompanied with the preferential adsorption of C_2H_6 over C_2H_4 at low-pressure regions. This is because the main binding sites for C_2H_6 and C_2H_4 are the same but with different binding strength, as revealed by our gas-loaded crystal structures (Figs. 4e and 4f). The full occupancy of these main binding sites corresponds to the saturated adsorption amount of 5.08 mmol g^{-1} . At 273 K, the C_2H_6 and C_2H_4 show quite similar uptakes of 5.0 and 5.13 mmol g^{-1} , both of which are consistent well with the saturated amount obtained from gas-loaded crystal structures. This indicates that the main binding sites are fully occupied for both C_2H_4 and C_2H_6 at 273 K. With the adsorption temperature increased to 296 K or 313 K, the adsorption occupancy of the main binding sites would be reduced for all C_2 gases; however, the stronger binding affinity of C_2H_6 results in a higher adsorption occupancy of the main binding sites than C_2H_4 (82% vs 68% at 296 K, and 59% vs 51% at 313 K), thus resulting in the preferential adsorption of C_2H_6 over C_2H_4 . This similar mechanism was also demonstrated by other materials for reversed C_2H_4/C_2H_6 separation (J. Am. Chem. Soc. 2020, 142, 633–640).

Therefore, the similar C_2H_6 and C_2H_4 uptakes at 273 K are mainly dominated by the full occupancy of the main binding sites for both gases. With the adsorption temperature increased to 296 K or 313 K, the stronger binding affinity of C_2H_6 enables a higher adsorption occupancy of the main binding sites than C_2H_4 , thus resulting in the more obviously preferential adsorption of C_2H_6 over C_2H_4 at 296 K and 313 K than at 273 K. To avoid the inappropriate description, we have revised the related sentence on page 10 highlighted in yellow, and the related explanations and more detailed discussion on effect of temperature have been included on page 3 and 4 in the revised Supplementary Information.

(4) Why does the Q_{st} for C_2H_6 show an initial increase with the uptake, followed by a decrease?

Response: Thank you very much for pointing out this issue. To address this issue, we carefully re-calculated all the Q_{st} curves of C_2H_2 , C_2H_6 and C_2H_4 for Al-PyDC based on better fitting of three adsorption isotherms at different temperatures. As shown in Fig. R1 or Supplementary Fig. 12, the re-calculated Q_{st} curve of C_2H_6 does not show the unusual behaviors, and exhibits a continuous increase in the whole range from 30.1 kJ mol^{-1} to 34.7 kJ mol^{-1} , which could be attributed to the intermolecular interactions among the C_2H_6 adsorbates during the adsorption process. This can be confirmed by our gas-loaded SCXRD study, in which the adsorbed C_2H_6 molecules show a close contact distance of 3.11–3.82 Å. The similar increase on C_2H_6 Q_{st} values was commonly observed in other C_2H_6 -selective MOFs such as MUF-15 due to the same reasons (J. Am. Chem. Soc. 2019, 141, 5014–5020). The new Q_{st} figure has been revised in Supplementary Fig. 12 accordingly.

Fig. R1 The recalculated adsorption enthalpies (Q_{st}) of C_2H_2 (red), C_2H_6 (blue) and C_2H_4 (black) for Al-PyDC.

(5) The disorder of C_2H_6 in the structure needs to be explained well. Also, the very short C6-C6 bond (1.09 Å) in C_2H_2 loaded structure needs to be checked.

Response: Thank you very much for your suggestion. In the process of crystal structure analysis for $C_2H_6@Al-PyDC$, we found that guest C_2H_6 molecules exhibited highly positional disorder within the channels. By atomic identification and refinement, we determined that the asymmetric unit of each guest C_2H_6 molecule within the channel includes four carbon atoms (C5, C6, C8, C9) and eight hydrogen atoms (H5A, H5C, H6A, H6C, H8A, H8C, H9A, H9C). Among them, four carbon atoms and four hydrogen atoms (H5A, H6A, H8A, H9A) are located on a mirror plane (-x, y, z), while two carbon atoms (C6 and C8) are located on a mirror plane (x, 1/2-y, z) and a 2-fold rotation axis (1/2-x, y, 1/2-z). The asymmetric unit of C_2H_6 is generated by these symmetry operations, resulting in four completely disordered C_2H_6 molecules at each active site, with each C_2H_6 molecule having an occupancy of 25%. The related explanations have been included on page 25 in the revised manuscript, highlighted in yellow.

Further, we refined the C_2H_2 -loaded crystal structure by using a more ideal C_2H_2 molecule model. The C6-C6 bond in the new C_2H_2 -loaded structure is 1.198 Å, which is now consistent well with the theoretical length of carbon-carbon triple bond (1.2 Å). It is noted that the adsorbed C_2H_2 molecules are still located at the same binding sites in the new structure, and the binding interactions between C_2H_2 and the framework remain almost unchanged with only very slight distance differences. This new C_2H_2 -loaded structures have been updated in CCDC, Fig. 4 in the revised manuscript and Table S4 in Supplementary Information. The related descriptions on binding interactions between C_2H_2 and the framework have also been revised on page 13 in the main text, highlight in yellow.

(6) The grammar needs improvement in some parts of the manuscript. For example:

Page 4, line 78: “the weakness nature” should be “the weak nature”

Page 4, lines 82-84: “UiO-67-(NH₂)₂ or CuTiF₆-TPPY holds the benchmark C₂H₂ and C₂H₆ uptakes or selectivities, but limited by inadequate gas selectivities or uptakes” is a bit confusing. Replacing “or” with “and” will make the sentence more clear.

Page 4, line 91: “no reports existed” should be “no reports existing”.

Page 7, line 148: “octahedrally AlO₆-polyhedras” should be “octahedral AlO₆-polyhedras”

Page 10, line 192: “It is worthy of note” should be “It is worthy to note”.

Response: Thank you very much for pointing out these issues. We have carefully checked and revised the grammatical issues throughout the manuscript accordingly. Thanks again for all your very constructive suggestions and comments, which helps us to improve the paper quality significantly.

Reviewer 2

Comments: This study reported that MOF-313 has promising properties for separating C₂H₄ from ternary mixtures of C₂H₂, C₂H₄, and C₂H₆. The scientific innovation and technological significance were found in engineering elements such as strong hydrocarbon absorption and corresponding hydrocarbon selectivity. They also exhibited chemical stability and ease of manufacture of the materials. On the other hand, previous studies have already reported MOF-313 production techniques and adsorption-based separation kinetics of C₂H₄ from mixtures. Thus, in my opinion, the overall novelty of this work is weak, and so more engineering data should be provided for publication in this journal. Here are some specific comments.

Response: Thank you very much for your comments and the following very constructive suggestions on more engineering data, which can certainly help us to improve the quality of this work significantly. Although the previous study has already reported MOF-313 (or called KMF-1) production techniques for water-sorption-driven cooling (at several tens of grams scale synthesized at 120 °C; Nat. Commun. 2020, 11, 5112), this is the first-ever use of this material as a robust and scalable adsorbent for benchmark one-step C₂H₄ purification from ternary C₂H₂/C₂H₄/C₂H₆ mixtures, one of the most challenging and important separation tasks at this stage. In addition, we also scaled up the green synthesis of Al-PyDC to multikilogram batches in an optimized reaction conditions with lower temperature (below water boiling point, 85 °C), without loss of any separation performance, which is more beneficial to implement large-scale applications and reduce production costs. More importantly, we here overcame the huge challenge to obtain large single crystals for this Al-MOF (note that the large-enough single crystal suitable for X-ray analysis was not realized before), enabling us to experimentally and visually identify the C₂ adsorption and separation mechanisms by single-crystal X-ray diffraction studies. Under the guidance of this reviewer on more engineering experiments, we have also included the engineering data including reported adsorption/desorption and breakthrough cycling experiments even under the presence of acid gases to assess the recyclability, stability and durability under actual circumstances. Based on these significances under the constructive guidance of this reviewer, we humbly think the revised manuscript in current version may have enough significance and importance to be published in this journal. Thank you again for your very constructive suggestions.

(1) A breakthrough experiment would be a useful indicator of adsorption-based mixture separation. It can, however, only describe one cycle of the adsorption process. This material should be evaluated in repeated adsorption and desorption environments, and related engineering data should be provided to improve technological significance of this work.

Response: Thank you very much for your very construction suggestion. We fully agree with the referee that the material should be evaluated in repeated adsorption and desorption environments to examine the recyclability. As suggested, we firstly carried out cycling experiments on Al-PyDC for single-component C_2H_2 and C_2H_6 adsorption. The adsorbent was exposed to C_2H_2 or C_2H_6 gas at 1 bar and 296 K until saturation, followed by desorption under vacuum at room temperature. As shown in Fig. R2 or Supplementary Fig. 18, Al-PyDC exhibits no obvious loss of C_2H_2 and C_2H_6 adsorption capacities over 30 repeated adsorption-desorption cycles.

Whereafter, we further performed dynamic breakthrough cycling measurements over 20 cycles for $C_2H_2/C_2H_6/C_2H_4$ (1/9/90) mixtures (Fig. R3, Fig. 5f or Supplementary Fig. 20) at 296 K and 1 bar, in which the sample was fully regenerated under sweeping He gas at 373 K. The almost unchanged retention time for C_2H_2 , C_2H_6 and C_2H_4 indicate that the separation performance of Al-PyDC can be preserved without any loss of performance over 20 continuous cycles. PXRD pattern and 77 K N_2 sorption isotherm after these repeated adsorption and breakthrough cycles indicate that Al-PyDC can maintain its structural integrity (Supplementary Fig. 21). These repeated adsorption and desorption cycling results indicate the technological significance of this material. The repeated breakthrough cycling figure has been included into Fig.5f and Supplementary Fig. 20, and all the related results have been added on page 16-17 in the main text, highlighted in yellow.

Fig. R2 The repeated adsorption-desorption cycles of Al-PyDC for C_2H_2 and C_2H_6 sorption at 296 K between the pressure of 1 bar and 0.001 bar.

Fig. R3 The twenty repeated adsorption-desorption cycles of breakthrough experiments on Al-PyDC for the 1/9/90 $C_2H_2/C_2H_6/C_2H_4$ mixture.

(2) Low temperature synthesis is one method of lowering production costs. It may, however, be associated with crystallinity deterioration and loss of selective function. The crystals formed in the low temperature procedure need to be tested for hydrocarbon separation.

Response: Thank you very much for your comments. We agree with the referee that low temperature synthesis might be associated with crystallinity deterioration and loss of selective function in some cases. To check this possibility in our large-scale sample, we first compared the SEM image of large-scale sample synthesized at low temperature (85 °C) with the sample produced at 120 °C (Fig. R4 or Supplementary Fig. 28). The SEM image showed that there are no obvious changes in their crystal morphology. Whereafter, we further measured the N_2 adsorption isotherms and C_2 adsorption isotherms for Al-PyDC samples synthesized at 85 °C, and compared them with the data obtained from ideal single-crystal sample. As shown in Fig. R5 or Supplementary Fig. 29, all the gas adsorption amounts show no obvious decrease compared to those of the crystal sample. The same situations were also obtained in the IAST selectivities and dynamic breakthrough experiments on Al-PyDC sample synthesized at low temperature. These results indicate that Al-PyDC sample synthesized at this low temperature can retain its crystallinity and high separation capacity. These related comments have been added on page 21 in the main text, highlighted in yellow.

Fig. R4 (left) The SEM images of large-scale sample synthesized at low temperature (85 °C), and (right) the sample synthesized at 120 °C according to the literature.

Fig. R5 Comparison of **a-c** C_2 gas adsorption at 296 K, **d** N_2 adsorption at 77 K, **e** IAST selectivity and **f** breakthrough experiment curves of kg-scale Al-PyDC sample synthesized at low temperature of 85 °C with those obtained from single crystal sample.

(3) I don't understand why aqueous environment was employed to confirm stability of the crystal. I think working environmental of the crystal toward the ternary mixture would be different, and may not include water vapors. It is necessary to assess the stability under actual or accelerated adsorption and desorption circumstances.

Response: Thank you very much for your very constructive suggestion. We fully agree with the referee that the actual working environmental would be more complex due to the presence of trace amount of acid gases (e.g., H_2S) in actual feed streams (*Fuel* 2019, 252, 553). To assess the stability of Al-PyDC sample under acid gas circumstances, we first carried out repeated adsorption/desorption cycling experiments for C_2H_6 gas containing 1000 ppm acidic H_2S gas,

wherein the sample was regenerated by N_2 sweeping at 373 K. As shown in Fig. R6 or Fig. 6c, it was demonstrated that the C_2H_6 adsorption capacity of Al-PyDC could be maintained with no noticeable loss after 50 cycles in the presence of acidic H_2S .

Next, the breakthrough experiments for 1/9/90 $C_2H_2/C_2H_6/C_2H_4$ mixtures containing 1000 ppm H_2S were performed on Al-PyDC sample to assess the effect of acidic gas on separation performance. As shown in Fig. R7 or Supplementary Fig. 22, the breakthrough times of C_2H_2 , C_2H_4 and C_2H_6 remain almost unchanged, affording a comparable C_2H_4 productivity of 1.57 mmol g^{-1} . More important, the separation capacity of Al-PyDC can be preserved without any loss of performance over six cycles under the presence of acid gases. PXRD pattern and 77 K N_2 sorption isotherm after such repeated breakthrough cycles under 1000 ppm H_2S conditions indicate that Al-PyDC can maintain its structural integrity (Supplementary Fig. 26). These results confirmed the high stability of Al-PyDC under actual circumstances. All these related comments have been added on pages 19 in the main text, highlighted in yellow.

Finally, given that an industrial application is potentially operated under higher gas flow rates, we thus conducted the dynamic breakthrough experiments for ternary $C_2H_2/C_2H_6/C_2H_4$ (1/9/90) mixtures under accelerated gas flow rates to evaluate the separation performance of Al-PyDC. As shown in Fig. R8 or Supplementary Fig. 17, with the flow rates increased to 5.0 and 10.0 mL min^{-1} , all the breakthrough times for C_2H_4 , C_2H_6 and C_2H_2 are accelerated significantly compared with those under 1.25 mL min^{-1} because of the larger gas flow to result in faster adsorption saturation. However, the clean separations were still retained, and the pure C_2H_4 productivities were calculated to be 1.50 and 1.45 mmol g^{-1} , respectively. This enables Al-PyDC to be a potential material for one-step C_2H_4 purification under industrial conditions. The related comments have been added in page 16, highlighted in yellow.

Fig. R6. C_2H_6 adsorption on Al-PyDC over 50 consecutive adsorption-desorption cycles for C_2H_6 gas containing 1000 ppm H_2S at 296 K, in which the sample was desorbed by N_2 sweeping at 120 °C.

Fig. R7 **a** Experimental breakthrough curves of Al-PyDC for 1/9/90 $C_2H_2/C_2H_6/C_2H_4$ mixture with 1000 ppm H_2S (red). **b** The cycling tests of Al-PyDC for 1/9/90 $C_2H_2/C_2H_6/C_2H_4$ mixture with 1000 ppm H_2S .

Fig. R8 Experimental column breakthrough curve for a 1/9/90 $C_2H_2/C_2H_6/C_2H_4$ mixture with a total flow of **a** 5 mL min^{-1} and **b** 10 mL min^{-1} in an absorber bed packed with Al-PyDC at 296 K and 1 bar.

REVIEWERS' COMMENTS

Reviewer #1 (Remarks to the Author):

The authors have carried out the required revisions, and the results obtained under acidic H₂S environment have made the manuscript stronger and elevated its quality. I can thus recommend it for publication without further revisions.

Reviewer #2 (Remarks to the Author):

I think the authors answered most of questions raised by reviewer. The work is now good to get acceptance.

Reviewer 1

Comments: The authors have carried out the required revisions, and the results obtained under acidic H₂S environment have made the manuscript stronger and elevated its quality. I can thus recommend it for publication without further revisions..

Response: Thank you very much for your very positive comments.

Reviewer 2

Comments: I think the authors answered most of questions raised by reviewer. The work is now good to get acceptance.

Response: Thank you very much for your comments and recommending the acceptance of our revised manuscript.